# RKKY Interaction in Graphene at Finite Temperature

**Eugene Kogan** [1,2] 

[1] Jack and Pearl Resnick Institute, Department of Physics, Bar-Ilan University, Ramat-Gan 52900, Israel; Eugene.Kogan@biu.ac.il

[2] Max-Planck-Institut fur Physik komplexer Systeme, 01187 Dresden, Germany

**Abstract:** In our publication from eight years ago (Kogan, E. 2011, vol. 84, p. 115119), we calculated Ruderman-Kittel-Kasuya-Yosida (RKKY) interaction between two magnetic impurities adsorbed on graphene at zero temperature. We show in this short paper that the approach based on Matsubara formalism and perturbation theory for the thermodynamic potential in the imaginary time and coordinate representation which was used then, can be easily generalized, and calculate RKKY interaction between the magnetic impurities at finite temperature.

**Keywords:** graphene; RKKY interaction; Matsubara Green's functions

## 1. Introduction

More than 60 years ago, it was understood that localized spins in metals can interact by means of the Ruderman-Kittel-Kasuya-Yosida (RKKY) mechanism [1–3]. This indirect exchange between two magnetic impurities in a non–magnetic host coupling is mediated by the conduction electrons and is traditionally calculated as the second order perturbation with respect to exchange interaction between the magnetic impurity and the itinerant electrons of the host. Though analysis of the RKKY interaction in the lowest order non-zero of perturbation is simple in principle, calculation of the integrals defining the interaction (whether analytical or numerical) can pose some problems.

RKKY interaction has been investigated in materials of different nature such as disordered metals [4], superconductors [5–7], topological insulators [8–15], carbon nanotubes [16,17], semiconducting wires [18], in Weyl and Dirac semimetals [19–24], but most thoroughly in graphene [25–46]. RKKY interaction in graphene is also the subject of the present publication.

The initial motivation for our study of the RKKY interaction in 2011 [47] was the realization of the fact that many of the previous calculations had to deal with divergent integrals, and the complicated (and to some extent arbitrary) cut-off procedure was implemented to obtain from these integrals the finite results. So we started to look for the procedure which will allow eliminating this problem. It turned out that by using Matsubara formalism [48] and calculating the Matsubara Green's functions in the coordinate-imaginary time representation, one is completely free from any diverging integrals in the whole calculation process [47].

Though Matsubara formalism is quite appropriate for finite temperature calculations (it was invented for that purpose) in our previous publications [35,47] we studied RKKY interaction only at zero temperature. The present short note is intended to generalize the result obtained previously to the case of finite temperature.

Adsorbed magnetic impurities can occupy different positions on graphene sheet. For example, they can be in the center of the elementary cell, or in between two adjacent graphene atoms [32,49–51]. However, in the previous publication we consider two magnetic impurities sitting on top of carbon atoms in graphene lattice, which is from our point of view the most interesting case. Let the two

impurities sit on top of the sites $i$ and $j$. Assuming a contact exchange interaction between the electrons and the magnetic impurities we can write the total Hamiltonian of the system as

$$H_T = H + H_{int} = H - J\mathbf{S}_i \cdot \mathbf{s}_i - J\mathbf{S}_j \cdot \mathbf{s}_j, \tag{1}$$

where $H$ is the Hamiltonian of the electron system, $\mathbf{S}_i$ is the spins of the impurity and $\mathbf{s}_i$ is the spin of itinerant electrons at site $i$.

## 2. Theoretical Methods

The consideration is based on the perturbation theory for the thermodynamic potential [48]. The correction to the thermodynamic potential due to interaction is

$$\Delta\Omega = -T\ln\langle S\rangle \equiv -T\ln\mathrm{tr}\left\{S \cdot e^{-H/T}/Z\right\}, \tag{2}$$

where the $S$–matrix is given by the equation

$$S = T_\tau \exp\left\{-\int_0^{1/T} H_{int}(\tau)d\tau\right\}. \tag{3}$$

Writing down $\mathbf{s}_i$ in the second quantization representation

$$\mathbf{s}_i = \frac{1}{2}c_{i\alpha}^\dagger \boldsymbol{\sigma}_{\alpha\beta} c_{i\beta}, \tag{4}$$

the second order term of the expansion with respect to the interaction is

$$\Delta\Omega = \frac{J^2 T}{4} \sum_{\alpha\beta\gamma\delta} \mathbf{S}_i \cdot \boldsymbol{\sigma}_{\alpha\beta} \mathbf{S}_j \cdot \boldsymbol{\sigma}_{\gamma\delta} \tag{5}$$

$$\int_0^{1/T}\int_0^{1/T} d\tau_1 d\tau_2 \left\langle T_\tau\left\{c_{i\alpha}^\dagger(\tau_1)c_{i\beta}(\tau_1)c_{j\gamma}^\dagger(\tau_2)c_{j\delta}(\tau_2)\right\}\right\rangle.$$

Please note that we have ignored the terms proportional to $\mathbf{S}_i^2$ and $\mathbf{S}_j^2$, because they are irrelevant for our calculation of the effective interaction between the adatoms spins. Actually, such and similar terms of higher order can potentially lead to the Kondo effect in graphene [52,53], and its competition with the RKKY interaction [54–56], but we do not touch this issue, implicitly assuming that the RKKY interaction is strong enough to suppress the Kondo effect (at the temperature considered).

Leaving aside the question about the spin structure of the two–particle Green's function standing in the r.h.s. of Equation (5) (for interacting electrons), further on we assume that the electrons are non–interacting. This will allow us to use Wick theorem and present the correlator from Equation (5) in the form

$$\left\langle T_\tau\left\{c_{i\alpha}^\dagger(\tau_1)c_{i\beta}(\tau_1)c_{j\gamma}^\dagger(\tau_2)c_{j\delta}(\tau_2)\right\}\right\rangle$$
$$= -\mathcal{G}_{\beta\gamma}(i,j;\tau_1-\tau_2)\mathcal{G}_{\delta\alpha}(j,i;\tau_2-\tau_1), \tag{6}$$

where

$$\mathcal{G}_{\beta\gamma}(i,j,\tau_1-\tau_2) = -\left\langle T_\tau\left\{c_{i\beta}(\tau_1)c_{j\gamma}^\dagger(\tau_2)\right\}\right\rangle \tag{7}$$

is the Matsubara Green's function [48]. We can connect $\mathcal{G}_{\beta\gamma}$ with the Green's function of spinless electron

$$\mathcal{G}_{\beta\gamma}(i,j,\tau_1-\tau_2) = -\delta_{\beta\gamma}\left\langle T_\tau\left\{c_i(\tau_1)c_j^\dagger(\tau_2)\right\}\right\rangle. \tag{8}$$

Presence of delta-symbols allows to perform summation with respect to spin indices in Equation (5)

$$\sum_{\alpha\beta} \mathbf{S}_i \cdot \sigma_{\alpha\beta} \mathbf{S}_j \cdot \sigma_{\beta\alpha} = \mathbf{S}_i \cdot \mathbf{S}_j, \tag{9}$$

which gives

$$\Delta\Omega = -J^2 \chi_{ij} \mathbf{S}_i \cdot \mathbf{S}_j, \tag{10}$$

where

$$\chi_{ij} = -\frac{1}{4} \int_0^{1/T} \mathcal{G}(i,j;\tau)\mathcal{G}(j,i;-\tau)d\tau \tag{11}$$

is the free electrons static real space spin susceptibility [47,57].

Thus we obtain

$$H_{RKKY} = -J^2 \chi_{ij} \mathbf{S}_i \cdot \mathbf{S}_j, \tag{12}$$

Green's function can be easily written down using representation of eigenvectors and eigenvalues of the operator $H$

$$(H - E_n) u_n = 0. \tag{13}$$

It is

$$\mathcal{G}(i,j;\tau) = \sum_n u_n^*(i) u_n(j) e^{-\xi_n \tau}$$
$$\times \begin{cases} -(1 - n_F(\xi_n)), & \tau > 0 \\ n_F(\xi_n), & \tau < 0 \end{cases}, \tag{14}$$

where $\xi_n = E_n - \mu$, and $n_F(\xi) = \left(e^{\beta\xi} + 1\right)^{-1}$ is the Fermi distribution function.

In calculations of the RKKY interaction in graphene the $\sum_n$ in Equation (14) turns into $\frac{a^2}{(2\pi)^2} \int d^2\mathbf{p}$, where $a$ is the carbon–carbon distance. (Actually, there should appear a numerical multiplier, connecting the area of the elementary cell with $a^2$, but we decided to discard it, which is equivalent to some numerical renormalization of $J$.) Also

$$u_n(i) = e^{i\mathbf{p}\cdot\mathbf{R}_i} \psi_{\mathbf{p}}, \tag{15}$$

where $\psi_{\mathbf{p}}$ is the appropriate component of spinor electron wave-function (depending upon which sublattice the magnetic adatom belongs to) in momentum representation.

Further in the integration with respect to $d^2\mathbf{p}$ we will treat as the integration in the vicinity of two Dirac points $K, K'$ and present $\mathbf{p} = \mathbf{K}(\mathbf{K}') + \mathbf{k}$. The wave function for the momentum around Dirac points $K$ and $K'$ has respectively the form

$$\psi_{\nu,\mathbf{K}}(\mathbf{k}) = \frac{1}{\sqrt{2}} \begin{pmatrix} e^{-i\theta_{\mathbf{k}}/2} \\ \nu e^{i\theta_{\mathbf{k}}/2} \end{pmatrix}$$
$$\psi_{\nu,\mathbf{K}'}(\mathbf{k}) = \frac{1}{\sqrt{2}} \begin{pmatrix} e^{i\theta_{\mathbf{k}}/2} \\ \nu e^{-i\theta_{\mathbf{k}}/2} \end{pmatrix}, \tag{16}$$

where $\nu = \pm 1$ corresponds to electron and hole band [58]; the upper line of the spinor refers to the sublattice $A$ and the lower line refers to the sublattice $B$.

In our publication from 2013 [35], we consider the case of doped graphene. However, here, like in our first publication on the subject [47], we consider only the case of undoped graphene, with the chemical potential at the Dirac points. The quantities $E_+(\mathbf{k})$ and $E_-(\mathbf{k})$ in this case would be electron and hole energy. Then, Equation (14) takes the form: for $i$ and $j$ belonging to the same sublattice

$$\mathcal{G}^{AA}(i,j;\tau > 0) = -\frac{1}{2}\left[e^{i\mathbf{K}\cdot\mathbf{R}_{ij}} + e^{i\mathbf{K}'\cdot\mathbf{R}_{ij}}\right]$$
$$\frac{a^2}{(2\pi)^2}\int d^2\mathbf{k}e^{i\mathbf{k}\cdot\mathbf{R}_{ij}-E_+(\mathbf{k})\tau}, \tag{17}$$

and for $i$ and $j$ belonging to different sublattices

$$\mathcal{G}^{AB}(i,j;\tau > 0) = \frac{1}{2}\frac{a^2}{(2\pi)^2}\int d^2\mathbf{k}e^{-E_+(\mathbf{k})\tau}$$
$$\times \left[e^{i(\mathbf{K}+\mathbf{k})\cdot\mathbf{R}_{ij}-i\theta_k} - e^{i(\mathbf{K}'+\mathbf{k})\cdot\mathbf{R}_{ij}+i\theta_k}\right]. \tag{18}$$

For $\tau < 0$ we should change the sign of the Green's functions and substitute $E_-$ for $E_+$.

For isotropic dispersion law $E(\mathbf{k}) = E(k)$ we can perform the angle integration in Equations (17) and (18) to get

$$\int d^2\mathbf{k}e^{i\mathbf{k}\cdot\mathbf{R}_{ij}-E(k)\tau} = \int_0^\infty dkkJ_0(kR)e^{-E(k)\tau} \tag{19}$$
$$\int d^2\mathbf{k}e^{i\mathbf{k}\cdot\mathbf{R}_{ij}\pm i\theta_k-E(k)\tau} = e^{\pm i\theta_\mathbf{R}}\int_0^\infty dkkJ_1(kR)e^{-E(k)\tau}$$

($J_0$ and $J_1$ are the Bessel function of zero and first order respectively, and $\theta_\mathbf{R}$ is the angle between the vectors $\mathbf{K} - \mathbf{K}'$ and $\mathbf{R}_{ij}$; $R = |\mathbf{R}_{ij}|$).

For the linear dispersion law

$$E_\pm(k) = \pm v_F k, \tag{20}$$

using mathematical identity [59]

$$\int_0^\infty x^{n-1}e^{-px}J_\nu(cx)dx \tag{21}$$
$$= (-1)^{n-1}c^{-\nu}\frac{\partial^{n-1}}{\partial p^{n-1}}\frac{\left(\sqrt{p^2+c^2}-p\right)^\nu}{\sqrt{p^2+c^2}},$$

we can explicitly calculate the Green's functions to get

$$\mathcal{G}^{AA}(i,j;\tau > 0) = -\frac{a^2}{4\pi}\frac{v\tau}{(v^2\tau^2+R^2)^{3/2}}$$
$$\left[e^{i\mathbf{K}\cdot\mathbf{R}_{ij}} + e^{i\mathbf{K}'\cdot\mathbf{R}_{ij}}\right] \tag{22}$$
$$\mathcal{G}^{AB}(i,j;\tau > 0) = \frac{a^2}{4\pi}\frac{R}{(v^2\tau^2+R^2)^{3/2}}$$
$$\left[e^{i(\mathbf{K}+\mathbf{k})\cdot\mathbf{R}_{ij}-i\theta_k} - e^{i(\mathbf{K}'+\mathbf{k})\cdot\mathbf{R}_{ij}+i\theta_k}\right]. \tag{23}$$

## 3. Results

Now we have substitute the results obtained into Equation (11). In our previous publication [47], only the case $T = 0$ was considered. However, consideration of finite temperature is no more complicated in the formalism used. (We again realize the convenience of the imaginary

time—coordinate representation of the Green's functions for the problem at hand. The transition from zero to finite temperature result can be performed just by changing limits of integration in the single integral.) Thus we obtain for arbitrary $T$

$$\chi_T^{AA}\left(\mathbf{R}_{ij}\right) = \chi_{T=0}^{AA}\left(\mathbf{R}_{ij}\right) \frac{16}{\pi} \int_0^{v/RT} \frac{x^2 dx}{(x^2+1)^3} \tag{24}$$

$$\chi_T^{AB}\left(\mathbf{R}_{ij}\right) = \chi_{T=0}^{AB}\left(\mathbf{R}_{ij}\right) \frac{16}{3\pi} \int_0^{v/RT} \frac{dx}{(x^2+1)^3}, \tag{25}$$

where $\chi_{T=0}^{AA}$ are the zero temperature susceptibilities calculated in our previous publication [47]

$$\chi_{T=0}^{AA}\left(\mathbf{R}_{ij}\right) = \frac{a^4}{256 v_F R^3}\left[1 + \cos((\mathbf{K}-\mathbf{K}')\cdot\mathbf{R}_{ij})\right] \tag{26}$$

$$\chi_{T=0}^{AB}\left(\mathbf{R}_{ij}\right) = -\frac{3a^4}{256 v_F R^3}\left[1 - \cos((\mathbf{K}-\mathbf{K}')\cdot\mathbf{R}_{ij} - 2\theta_\mathbf{R})\right]. \tag{27}$$

Actually, while rederiving Equations (28) and (29) we have found additional multiplier $1/2\pi$, but since we were already quite sloppy with the numerical multiplier in going from summation to integration in Equation (14), we decided to leave the equations in this paper as they were in the previously published one [47]. In any case, Equations (24) and (26) are valid independently of the presence of additional multiplier for $\chi_{T=0}$.

Integrals in Equations (24) and (26) can be easily calculated, and we obtain for the intrasublattice interaction

$$\chi_T^{AA}\left(\mathbf{R}_{ij}\right) = \chi_{T=0}^{AA}\left(\mathbf{R}_{ij}\right)$$
$$\frac{2}{\pi}\left[\tan^{-1}z + \frac{z}{z^2+1} - \frac{2z}{(z^2+1)^2}\right], \tag{28}$$

and for the intersublattice interaction

$$\chi_T^{AB}\left(\mathbf{R}_{ij}\right) = \chi_{T=0}^{AB}\left(\mathbf{R}_{ij}\right)$$
$$\frac{2}{\pi}\left[\tan^{-1}z + \frac{z}{z^2+1} + \frac{z}{3(z^2+1)^2}\right], \tag{29}$$

where $z = v/R\tau$. The limiting cases of Equations (30) and (31) are particularly simple. For $T \ll v/R$ the first term in the braces both in Equation (30) and in Equation (31) is equal to $\pi/2$ and the other two terms can be neglected. Thus we obtain the previous ($T = 0$) results. The opposite limiting case $T \gg v/R$ is easier to get directly from Equations (24) and (26). Expanding the integrand we get for $T \gg v/R$

$$\chi_T^{AA}\left(\mathbf{R}_{ij}\right) = \chi_{T=0}^{AA}\left(\mathbf{R}_{ij}\right) \frac{16}{3\pi}\left(\frac{v}{RT}\right)^3 \tag{30}$$

$$\chi_T^{AB}\left(\mathbf{R}_{ij}\right) = \chi_{T=0}^{AB}\left(\mathbf{R}_{ij}\right) \frac{16}{3\pi}\frac{v}{RT}. \tag{31}$$

We must mention that comparing our results with those obtained earlier for the case of doped graphene [60], one should be aware of the fact that the exponential decrease of the RKKY interaction with the distance at high temperatures obtained in Ref. [60], was obtained for $k_F R \gg 1$ (in our case $k_F = 0$).



## 4. Conclusions

We have calculated RKKY interaction between two magnetic impurities adsorbed on graphene at arbitrary temperature. Matsubara Green's functions in coordinate and imaginary time representations were used. For zero temperature the results coinsides with those obtained by us previously. At high temperature the interaction decreases inversely proportional to to the temperature for the inter-sublattice interaction, and inversely proportional to the third power of temperature fore the intra-sublattice interaction.

**Funding:** This research received no external funding.

**Acknowledgments:** This paper was written during the author's visit to Max-Planck-Institut fur Physik komplexer Systeme in 2019. The author cordially thanks the Institute for the hospitality extended to him during that and all the previous visits.

**Conflicts of Interest:** The authors declare no conflict of interest.

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
