# Peer review of "RKKY Interaction in Graphene at Finite Temperature"

_carbon, 2019_

Round 1
Reviewer 1 Report
In the manuscript “RKKY interaction in graphene at finite temperature”, the author extended the method for calculating RKKY interaction with finite temperatures. It is an interesting manuscript. However, I have some concerns.
P3 “But consideration of finite temperature is no more complicated in the formalism used. Thus we obtain for arbitrary T ...eqs (24) and (25)”. Please provide more detailed information for “how to obtain the eqs (24) and (25)”
Please provide 1-2 Figures for the examples with finite temperatures and compared them with the one with T = 0.
P1 “Adsorbed magnetic impurities can occupy different positions on graphene sheet.” References (e.g. Energy storage materials, 2019, 16, 619-624; Journal of Materials Chemistry A, 2018, 6, 6815-6821; Journal of physical chemistry letters, 2019, 10, 721-726) are needed to support this statement.
Author Response
I am grateful to the Referee for the appreciation of the paper and for the constructive critical remarks. The following changes have been made in the manuscript in response to the latter.
The paragraph was added before Eqs. (24) and (25) to explain there derivation.
The RKKY interaction was explicitly calculated for arbitrary temperature (Eqs. (28) and (29)), so the relation between the zero and finite temperature results is now obvious. Also, the transition to the limiting cases of low and high temperature is explained in more details.
The references supporting the statement about possible positions of adatoms in the graphene lattice were added to the paper.
We hope that the new version corresponds to the standards of the Journal.
Reviewer 2 Report
This short note is an extension of the work reported elsewhere by the author. In the reported work, the author had calculated the RKKY interaction between two magnetic impurities adsorbed on graphene at zero temperature. The current manuscript is an extension of that work to finite temperatures. Although a brief extension of the previous work, the increment in the scientific understanding provided is suitable for publication in 'C' as a short note. I recommend accepting the manuscript for publication in its current form.
Author Response
I am very grateful to the Referee for the appreciation of the paper.
Following his/her remarks the spell check was performed and the style was improved.
We hope that the new version corresponds to the standards of the Journal.